# Using computed tomography to evaluate proper chest compression depth for cardiopulmonary resuscitation in Thai population: A retrospective cross-sectional study

Pongsakorn Atiksawedparit[1], Thanaporn Sathapornthanasin[2], Phanorn Chalermdamrichai[2], Pitsucha Sanguanwit[2], Nitima Saksobhavivat[ID][3], Ratchanee Saelee[4], Phatthranit Phattharapornjaroen[ID][2]*

1 Faculty of Medicine Ramathibodi Hospital, Chakri Naruebodindra Medical Institute, Mahidol University, Bangkok, Thailand, 2 Faculty of Medicine Ramathibodi Hospital, Department of Emergency Medicine, Mahidol University, Bangkok, Thailand, 3 Faculty of Medicine Ramathibodi Hospital, Department of Diagnostic and Therapeutic Radiology, Mahidol University, Bangkok, Thailand, 4 Faculty of Medicine Ramathibodi Hospital, Department of Internal Medicine, Mahidol University, Bangkok, Thailand

* phatthranit.pha@mahidol.edu

## Abstract

### Introduction

The effectiveness of cardiopulmonary resuscitation is determined by appropriate chest compression depth and rate. The American Heart Association recommended CC depth at 5–6 cm to indicate proper cardiac output during cardiac arrest. However, many studies showed the differences in the body builds between Caucasians and Asians. Therefore, this study aimed to determine heart compression fraction (HCF) in the Thai population by using contrast-enhanced computed tomography (CT) scan of the chest and a mathematical model.

### Materials and methods

Consecutive contrast-enhanced CT scans of the chest performed at Ramathibodi Hospital were retrospectively reviewed from January to March 2018 by two independent radiologists. Patients' characteristics, including gender, age, weight, height, and pre-existing diseases, were recorded, and the chest parameters were measured from a CT scan. The heart compression fraction (HCF) was subsequently calculated.

### Results

Of 306 subjects, there were 139 (45.4%) males, 148 (47.4%) lung diseases and 10 (3.3%) heart diseases. Mean age and BMI were 60.4 years old and 23.8 kg/m$^2$, respectively. Chest diameter, heart diameter, and non-cardiac soft tissue were significantly smaller in females compared to males. Mean (SD) HCF proportional with 50 mm and 60 mm depth were 38.3% (13.3%) and 50% (14.3%), respectively. There were significant differences of HCF

**Data Availability Statement:** All relevant data are within the paper and its Supporting Information files.

**Funding:** The authors received no specific funding for this work.

**Competing interests:** The authors have declared that no competing interests exist.

proportional by 50 mm and 60 mm depth between men and women (33.2% vs 42.6% and 44% vs 54.9%, respectively (P<0.001)). In addition, a decrease in HCF was significantly observed among higher BMI groups.

## Conclusion

The CT scan and mathematical model showed that 38% and 50% HCF proportions were generated by 50 mm and 60 mm CC depth. HCF proportions were significantly different between genders and among BMI groups. The recommended depth of 5–6 cm is likely to provide sufficient CC depth in the population of Thailand.

## Introduction

Closed-chest compression (CC), first introduced in 1960 [1], increases coronary perfusion pressure and generates cardiac output enhancing the chances of survival for patients with cardiac arrest. The term 'Chain of Survival' is a series of critical actions that improve survival following cardiac arrest. The six interdependent links in the chain of survival consist of early recognition of cardiac arrest and activation of the emergency team, early cardiopulmonary resuscitation (CPR), rapid defibrillation, early advanced cardiac life support, advanced resuscitation by emergency medical services, post-cardiac arrest care, and recovery [2,3]. Previous findings suggested that the appropriate external CC during CPR improves high survival rates and favorable neurologic outcomes for patients with cardiac arrest [4,5]. External CC generates forward blood flow of approximately one-third of the normal cardiac output [6,7] by increasing intrathoracic pressure (thoracic pump theory) and directly squeezing the heart (cardiac pump theory) [8–10]. According to the cardiac pump theory, direct ventricular compression significantly results in cardiac output.

During CPR, the effectiveness of CC depends on compression rate and depth [11]. To generate adequate cardiac output and prevent complications of CC such as liver or stomach rupture, European Resuscitation Council guidelines 2020 and American Heart Association 2020 recommended that, irrespective of rescuers variables, CC depth should be at least 5 cm, but not greater than 6 cm [12]. According to cardiac pump theory, this depth can generate 25%–30% of average cardiac output among standard adults [5,13,14]. Currently, computed tomography (CT) scan with contrast of the chest was used to simulate heart compression fraction (HCF), which is the proportion of the heart compressed by CC [15]. The HCF was mentioned in several studies to demonstrate the cardiac output in relation to the chest compression depth [16–18]. Higher HCF reflects greater cardiac output to the vital organ during chest compression, which increases the quality of CPR and the chance of survival.

Although 50 mm to 60 mm chest compression depth was recommended regardless of gender, body size, and medical history, many studies showed the effects of age and body structure on the HCF irrespective of rescuers' factors [17,18]. Moreover, higher BMI patients significantly presented higher chest compartment diameters despite their lower HCF [15,19,20]. In addition, some evidence suggested that 50 mm to 60 mm chest compression depth generated inadequate HCF among geriatric patients [21]. In contrast, previous research recommended performing deeper CC in special populations, which consequently reported complications such as sternal/rib fractures, flail chest, pneumohemothorax, and liver laceration [22–26]. Considering the differences in body structures across ethnicities, it can be inferred that the most suitable CC depth might not always follow the standard recommendation [27].

Therefore, this study aimed to determine heart compression by CC, particularly HCF, among the Thai population using a CT scan and a mathematical model using contrast-enhanced computed tomography (CT) scan of the chest.

## Materials and methods

The study was approved by the ethics committee of the Faculty of Medicine, Ramathibodi Hospital, Mahidol University, under approval number COA. MURA2019/1061 on Oct 28, 2019. As this study used secondary data, obtaining informed consent was waived by the ethics committee of the Faculty of Medicine, Ramathibodi Hospital, Mahidol University.

### Study design and setting

A retrospective cross-sectional study was performed at Ramathibodi Hospital, an 800- bed capacity medical school. All consecutive Thai adults (18 years or older) who underwent CT chest with contrast from January 2018 to March 2018 were enrolled. Patients with anatomical abnormalities (i.e., pectus carinatum and excavatum, trauma-related deformities, dextrocardia, situs inversus, and mediastinal shift or patients with the absence of nipple were excluded. Their medical records and CT scans with a chest contrast were reviewed. The sample size was calculated based on the estimation of the mean by using the standard deviations of internal anteroposterior diameter (IAPD) and external anteroposterior diameter (EAPD) in chest CT which were 10.79 (±1.48) cm. and 22 (±2.13) cm., 0.05 α- error, 0.2 ß- error, and a missing rate of 10% into account [19]. Therefore 306 subjects were recruited for this study.

### Data collection

Data were divided into general characteristics and chest compartments domain. General characteristic domains, i.e., age, gender, weight, height, body mass index (BMI), and hometown, were extracted from medical records. According to the World Health Organization guideline, BMI was classified into four categories: underweight ($<18.5$ kg/m$^2$), ideal body weight (18.5–24.9 kg/m$^2$), overweight (25.0–29.9 kg/m$^2$), and obese ($\geq30$ kg/m$^2$) [28–30]. Chest compartments domain includes external chest anteroposterior (AP) diameter (mm), which is the distance measured at the midline at the axial level of maximal left ventricular diameter (LV max) from the skin overlying the sternum to the skin posterior to the spinous process on the back, internal chest AP diameter (mm), which is the midline distance from the posterior cortex of the sternum to the anterior cortex of the vertebral body; and heart AP diameter (mm), which is the midline distance from anterior to posterior walls of the heart (Fig 1). Non-cardiac soft tissue is estimated by subtracting cardiac AP diameter from internal chest AP diameter. The HCF was estimated by the following equation [15];

$$HCF = \frac{X - d}{heart\ AP\ diameter} \times 100$$

*Where*:
   *X: proposed CC depths, which were 50 mm or 60 mm*
   *d: noncardiac thoracic tissue*

### Statistical analysis

Sample size calculation by estimating mean formula, using mean and standard deviation (SD) of chest IAPD [19] determined that 306 subjects were required for our study. Data were compiled using an Excel spreadsheet (Excel 2010; Microsoft, Redmond, WA, USA) and analyzed

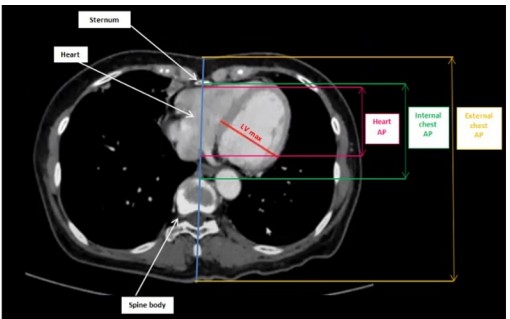

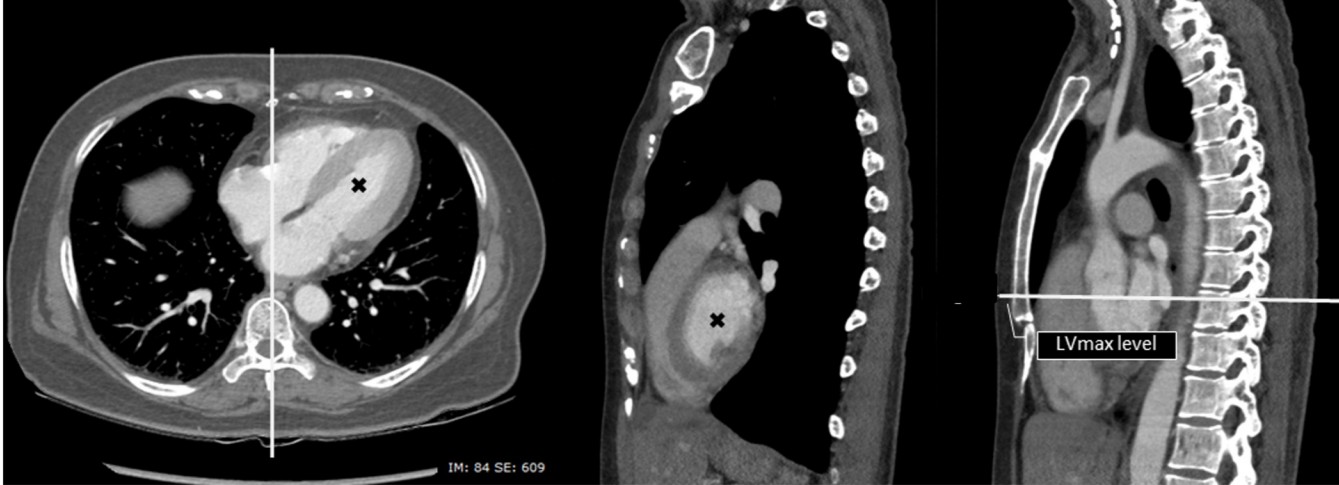

**Fig 1. Axial chest CT image at the level of maximal left ventricular diameter (LV max).** External chest anteroposterior diameter, internal chest anteroposterior diameter, and anteroposterior heart diameter were measured in the midline as shown. **B**, Midline sagittal chest CT image. The transverse white line represents the level of LV max demonstrated on the axial view, which corresponds to the mid-lower half of the sternum in this subject. (AP, anteroposterior. LV max, maximal left ventricular diameter on axial plane).

using STATA version 15. The categorical variables are shown in number and percentage. In contrast, continuous variables were described as mean with SD or median with interquartile range (IQR) according to normal or non-normal distribution. The student t-test or Mann-Whitney U-test compared two independents, normally or non-normally distributed continuous data. More than 2 groups' comparison of means was determined by Analysis of Variance (ANOVA). In addition, Analysis of Covariance (ANCOVA) was used to adjust age and sex, which might affect the association of BMI and outcomes. Univariate and multivariate quartile regressions were used to compare more than two groups non normally distributed continuous data. The chi-square test or Fisher exact test was used to compare categorical data. Statistical significance was achieved if the P-value was less than 0.05. Statistical analysis was performed by using the STATA 15 software.

## Results

Of 417 eligible subjects, 111 were excluded according to exclusion criteria, and 306 patients were included in the final analysis (Fig 2). The baseline characteristics of the subjects are displayed in Table 1. There were 139 (45.4%) male, 148 (47.4%) lung diseases and 10 (3.3%) heart diseases. Mean (SD) age and BMI were 60.4 (12.7) years old and 23.8 (4.2) kg/m$^2$, respectively. Half of the subjects were in 18.5–24.9 kg/m2 BMI groups, and their hometowns were in the middle region of Thailand. There was significantly higher weight (66.9 kg VS 56.8 kg,

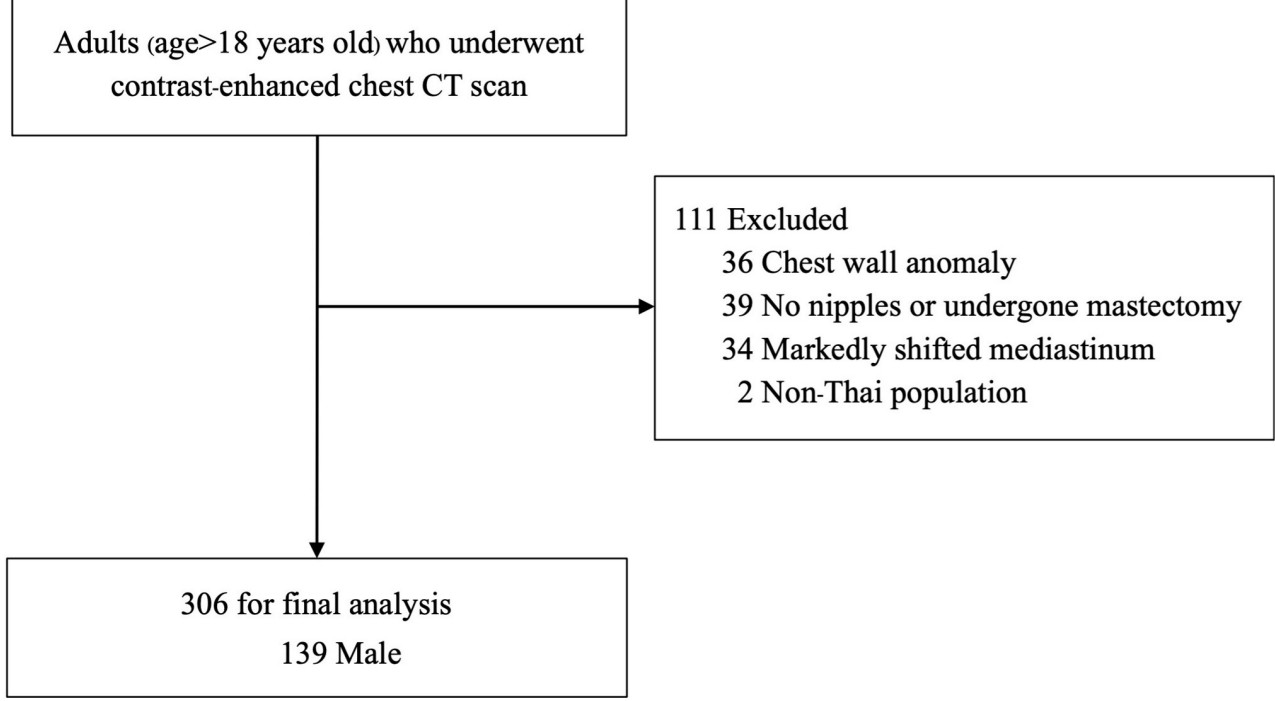

**Fig 2. Subject selection.**

P < 0.001) and height (166.2 (6.1) cm. VS 155.1 (5.8) cm., P<0.001) of males compared to females. No significant difference was observed in age, BMI, regions, and underlying diseases between genders.

The mean (SD) of external and internal chest diameters was 216.2 (25.1) mm and 105.2 (17.4) mm, respectively. The mean heart diameter was 88.1 (14.3) mm, whereas the median (IQR) of non-cardiac soft tissue diameter was 16.3 (9.9, 23.7) mm. There was a significant difference in external chest diameter, internal chest diameter, and heart diameter between genders (Table 2).

The proportion of chest compartments and chest compression depth to external chest diameter was described in Table 3. The proportion of internal chest and heart diameters to external chest diameter was bigger in males than females, yet females' 50mm and 60mm CC depth was bigger. HCF, generated by 50 mm and 60 mm CC depth, was 38.3% and 50%, respectively. Compared to males, higher HCF by 50 mm and 60 mm depth of females were observed (42.6% VS 33.2% and 54.9% VS 44%, P < 0.001).

Chest compartments—the proportion of chest compartments to external chest diameter-and HCF, based on BMI, were displayed in Table 4. Although larger internal chest and heart diameters were observed among higher BMI groups, the proportion to external chest compression was not significantly different among BMI groups. Moreover, we also found that HCF by 50 mm and 60 mm CC depth were inversely proportional to BMI groups. These significant findings were not changed after adjusting for age and sex.

## Discussion

We have determined the HCF of CC based on CT scan findings among the Thai population, and the results revealed that CC by 50 mm to 60 mm depth yielded HCF of 38.3% to 50%,

**Table 1. General characteristics of included subjects.**

| Characteristics | Total (N = 306) | Gender | | P value |
|---|---|---|---|---|
| | | Male (n = 139) | Female (n = 167) | |
| Age (years), mean (SD) | 60.4 (12.7) | 62.4 (11.3) | 58.8 (13.5) | 0.015 |
| Weight (kg), mean (SD) | 61.4 (12.5) | 66.9 (12.3) | 56.8 (10.7) | <0.001 |
| Height (cm), mean (SD) | 160.1 (8.1) | 166.2 (6.1) | 155.1 (5.8) | <0.001 |
| BMI (kg/m$^2$), mean (SD) | 23.8 (4.2) | 24.1 (3.9) | 23.6 (4.4) | 0.295 |
| BMI groups, n (%) | | | | |
| Underweight | 27 (8.82) | 6 (4.32) | 21 (12.57) | 0.036 |
| Normal | 167 (54.58) | 77 (55.4) | 90 (53.89) | |
| Overweight | 84 (27.45) | 45 (32.37) | 39 (23.35) | |
| Obese | 28 (9.15) | 11 (7.91) | 17 (10.18) | |
| Regions, n (%) | | | | |
| Middle | 176 (57.5) | 81 (58.3) | 95 (56.9) | 0.085 |
| West | 19 (6.2) | 7 (5) | 12 (7.2) | |
| East | 22 (7.2) | 9 (6.5) | 13 (7.8) | |
| North East | 46 (15) | 28 (20.1) | 18 (10.8) | |
| North | 8 (2.6) | 1 (0.7) | 7 (4.2) | |
| South | 35 (11.4) | 13 (9.4) | 22 (13.2) | |
| Lung diseases, n (%) | | | | |
| None | 161 (52.6) | 77 (55.4) | 84 (50.3) | 0.635 |
| COPD/Asthma | 18 (5.9) | 9 (6.5) | 9 (5.4) | |
| cancer/metastasis | 112 (36.6) | 48 (34.5) | 64 (38.3) | |
| Tuberculosis | 15 (4.9) | 5 (3.6) | 10 (6) | |
| Heart diseases, n (%) | | | | |
| None | 296 (96.7) | 135 (97.1) | 161 (96.4) | 1.00 |
| IHD/arrhythmia | 10 (3.3) | 4 (2.9) | 6 (3.6) | |

BMI indicates body mass index, cm; centimeter, COPD; chronic obstructive pulmonary disease, kg; kilogram, and IHD; ischemic heart disease.

respectively. Significantly higher HCF was found among females compared to males. Additionally, there was a decrease in HCF among higher BMI groups.

The characteristics of our subjects were elderly and normal to overweight BMI groups, which were not different from the subjects reported in the previous studies in Asia [15,19,21]. Although 47.4% of our subjects had lung diseases, which were not reported in the earlier studies, external and internal chest, as well as heart diameter, was not clinically different (216.2 mm VS 220 mm and 105.2 mm VS 107.9 mm) [14]. Besides, higher chest compartment diameters were found among higher BMI groups which were similar to previous findings [13,14].

**Table 2. Size of each chest compartments.**

| Chest compartments (mm) Mean (SD) | Total (N = 306) | Gender | | P value |
|---|---|---|---|---|
| | | Male (n = 139) | Female (n = 167) | |
| External chest diameter | 216.2 (25.1) | 226.6 (24.3) | 207.6 (22.4) | <0.001 |
| Internal chest diameter | 105.2 (17.4) | 113.5 (16.2) | 98.2 (15.2) | <0.001 |
| Heart diameter | 88.1 (14.3) | 94.4 (13.5) | 82.8 (12.7) | <0.001 |
| Non-cardiac soft tissue: Median (IQR) | 16.3 (9.9, 23.7) | 17.3 (11.6, 25.5) | 14.9 (9.1, 21) | 0.006 |

mm indicates millimeter, IQR: Interquartile range and SD; standard deviation.

**Table 3. HCF and proportion of chest compartments to external chest diameter, based on gender.**

| Variable | Total (N = 306) Mean (SD) | Gender | | P value |
|---|---|---|---|---|
| | | Male (n = 139) | Female (n = 167) | |
| **Proportion to external chest diameter (%)** | | | | |
| Internal chest diameter | 48.4 (3.9) | 49.9 (3.4) | 47.2 (3.9) | <0.001 |
| Heart diameter | 40.6 (3.8) | 41.6 (3.6) | 39.8 (3.7) | <0.001 |
| 50 mm CC depth | 23.4 (2.7) | 22.3 (2.4) | 24.3 (2.6) | <0.001 |
| 60 mm CC depth | 28.1 (3.3) | 26.7 (2.9) | 29.2 (3.2) | <0.001 |
| **HCF (%)** | | | | |
| By 50 mm CC depth | 38.3 (13.3) | 33.2 (12.3) | 42.6 (12.5) | <0.001 |
| By 60 mm CC depth | 50 (14.3) | 44 (12.9) | 54.9 (13.6) | <0.001 |

CC indicates chest compression, HCF; heart compression fraction, mm; millimeter and SD: Standard deviation.

Therefore, the smaller chest compartment size of females compared to males might be explained by lower BMI.

Based on the cardiac pump model, direct CC mainly generated approximately 33% of the ejection fraction in normal adults, i.e., EF of 67% in 70 kilograms of healthy people [6,7]. The results of mean HCF from 38.3% to 50% were estimated by 50 mm and 60 mm CC depth and were consistent with other findings in Asia, which ranged from 37.1% to 54.6% [15,19,21]. Therefore, 50 mm to 60 mm CC depths might generate sufficient cardiac output during CPR.

Our results indicated that lower HCF was found among higher BMI groups which were consistent with the previous findings [15,19,21]. This might be explained by obese patients having a wider thoracic compartment and larger non-cardiac soft tissue, whereas CC depth was fixed at 50 mm to 60 mm. Although lower HCF was found in the obese groups, sufficient HCF was generated at 60 mm CC depth. However, the appropriate CC depth among obese

**Table 4. Chest compartments, the proportion of chest compartments to external chest diameter and HCF, based on BMI indicates body mass index, CC; chest compression, HCF; heart compression fraction, mm; millimeter and SD: Standard deviation.**

| Variables Mean (SD) | BMI groups | | | | P value | Adjusted P value* |
|---|---|---|---|---|---|---|
| | Underweight | Normal | Overweight | Obese | | |
| | n = 27 | n = 167 | n = 84 | n = 28 | | |
| **Chest compartments (mm)** | | | | | | |
| External chest diameter | 179.8 (11.7) | 207.7 (17.1) | 232.9 (16.5) | 252.8 (19.2) | <0.001 | <0.001 |
| Internal chest diameter | 85.7 (10.8) | 100.7 (14.3) | 113.9 (14.1) | 124.6 (17.6) | <0.001 | <0.001 |
| Heart diameter | 73.7 (9.2) | 83.5 (11.7) | 96.9 (11.7) | 103.6 (14.6) | <0.001 | <0.001 |
| Non-cardiac soft tissue diameter (Median (IQR)) | 12.2 (7.2, 17.6) | 16.8 (9.9, 24.2) | 15.5 (10.2, 21.73) | 19.4 (16.2, 27.9) | 0.008 | 0.036 |
| **Proportion to external chest diameter (%)** | | | | | | |
| Internal chest diameter | 47.5 (3.6) | 48.3 (4) | 48.7 (3.4) | 49.1 (4.7) | 0.401 | 0.620 |
| Heart diameter | 40.6 (3.9) | 40.1 (3.6) | 41.6 (3.7) | 40.9 (4.3) | 0.040 | 0.030 |
| 50 mm CC depth | 27.9 (1.8) | 24.2 (2.6) | 21.5 (1.5) | 19.8 (1.5) | <0.001 | <0.001 |
| 60 mm CC depth | 33.5 (2.1) | 29 (2.4) | 25.8 (1.8) | 23.8 (1.8) | <0.001 | <0.001 |
| **HCF (%)** | | | | | | |
| By 50 mm CC depth | 51.6 (11.7) | 39.8 (13.3) | 34.4 (10.6) | 28.7 (10.8) | <0.001 | <0.001 |
| By 60 mm CC depth | 65.5 (12.5) | 52 (13.9) | 44.8 (11.6) | 38.6 (10.9) | <0.001 | <0.001 |

BMI indicates body mass index, CC; chest compression, HCF; heart compression fraction, mm; millimeter and SD: Standard deviation.

*Adjusted for age and sex.

cardiac arrest patients requires further studies to tradeoff between proper cardiac output and complications such as liver rupture, rib fractures, sternal fractures, etc.

Our study had some limitations. First, we estimated HCF based on cardiac pump theory and did not consider thoracic pump theory. However, HCF is the minimum blood flow generated during CC, and it is enough to determine the lower threshold of stroke volume during CPR. Second, this study contained some selection biases because the participants were consecutively selected from a list of cases referred to our center which were assumed to have complex respiratory conditions, 47.4% of lung diseases, whereas a small number of patients with a higher risk of cardiac arrest, 3.3% of cardiac illnesses, were enrolled. Although this selection bias may influence the application of the results in a target population, lung and heart diseases themselves may not be the significant factor affecting anatomical parameters and HCF. Regarding different body sizes of the population (across regions) in Thailand, a multicenter study might be required to represent the overall population. Regarding different body sizes of the population (across regions) in Thailand, a multicenter study might be required to represent the overall population.

## Conclusion

Our result revealed that CC by 50 mm to 60 mm depth yielded HCF of 38.3% to 50%, respectively, sufficient to generate stroke volume. Nonetheless, significantly higher HCF was found among females compared to males. There was a decrease in HCF among higher BMI groups based on 50 mm to 60 mm CC depth; however, sufficient HCF was generated at 60 mm CC depth in all BMI groups.

## Supporting information

**S1 File.**
(XLS)

## Author Contributions

**Conceptualization:** Pongsakorn Atiksawedparit, Phanorn Chalermdamrichai, Pitsucha Sanguanwit, Nitima Saksobhavivat, Ratchanee Saelee, Phatthranit Phattharapornjaroen.

**Data curation:** Thanaporn Sathapornthanasin, Phanorn Chalermdamrichai, Pitsucha Sanguanwit, Nitima Saksobhavivat, Ratchanee Saelee, Phatthranit Phattharapornjaroen.

**Formal analysis:** Pongsakorn Atiksawedparit, Pitsucha Sanguanwit, Nitima Saksobhavivat, Ratchanee Saelee, Phatthranit Phattharapornjaroen.

**Investigation:** Thanaporn Sathapornthanasin, Phanorn Chalermdamrichai, Nitima Saksobhavivat, Phatthranit Phattharapornjaroen.

**Methodology:** Pongsakorn Atiksawedparit, Thanaporn Sathapornthanasin, Pitsucha Sanguanwit, Nitima Saksobhavivat, Ratchanee Saelee, Phatthranit Phattharapornjaroen.

**Project administration:** Phatthranit Phattharapornjaroen.

**Resources:** Phatthranit Phattharapornjaroen.

**Supervision:** Pongsakorn Atiksawedparit, Phanorn Chalermdamrichai, Phatthranit Phattharapornjaroen.

**Validation:** Pongsakorn Atiksawedparit, Thanaporn Sathapornthanasin, Phatthranit Phattharapornjaroen.

**Visualization:** Pongsakorn Atiksawedparit, Thanaporn Sathapornthanasin, Phatthranit Phattharapornjaroen.

**Writing – original draft:** Thanaporn Sathapornthanasin, Phatthranit Phattharapornjaroen.

**Writing – review & editing:** Pongsakorn Atiksawedparit, Thanaporn Sathapornthanasin, Phanorn Chalermdamrichai, Pitsucha Sanguanwit, Nitima Saksobhavivat, Ratchanee Saelee, Phatthranit Phattharapornjaroen.

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
