## [Decision Letter · Decision Letter 0]

15 Jul 2022

PONE-D-22-06863Using Computed Tomography to Evaluate Proper Chest Compression Depth for Cardiopulmonary Resuscitation in Thai Population: A Retrospective Cross-Sectional StudyPLOS ONE

Dear Dr. Phattharapornjaroen,

Thank you for submitting your manuscript to PLOS ONE. After careful consideration, we feel that it has merit but does not fully meet PLOS ONE’s publication criteria as it currently stands. Therefore, we invite you to submit a revised version of the manuscript that addresses the points raised during the review process.

ACADEMIC EDITOR: Thank you very much for having submitted your paper for consideration.  Some issues have been raised by the reviewers. Please follow their indication to try and improve your manuscript

We look forward to receiving your revised manuscript.

Kind regards,

Simone Savastano

Academic Editor

PLOS ONE

Additional Editor Comments:

Thank you very much for having submitted your paper for consideration. Some issues have been raised by the reviewers. Please follow their indication to try and improve your manuscript

Reviewers' comments:

Reviewer's Responses to Questions

**Comments to the Author**

1. Is the manuscript technically sound, and do the data support the conclusions?

Reviewer #1: Partly

Reviewer #2: Yes

2. Has the statistical analysis been performed appropriately and rigorously? 

Reviewer #1: No

Reviewer #2: Yes

3. Have the authors made all data underlying the findings in their manuscript fully available?

Reviewer #1: No

Reviewer #2: Yes

4. Is the manuscript presented in an intelligible fashion and written in standard English?

Reviewer #1: Yes

Reviewer #2: Yes

5. Review Comments to the Author

Reviewer #1: The depth of chest compressions depends on the rescuer's BMI anywhere in the world. My brother-in-law (Thai nurse) weighs 78 kg and 174 centimeters tall, he certainly has no problem make chest compression 5 -6 centimeters deep a European patient. Much depends on the difference in weight between the rescuer and the victim also in the Caucasians and

Asians populations.

Reviewer #2: This is an interesting retrospective study regarding the distribution of heart compression fraction among different sex and BMI in Thai population. The conclusion is that the recommended depth of chest compression during cardiac arrest (5-6 cm) is validated (with the simulated chest ct scan model in object) also in Thai population.

The article is well written in sound English, easy to read, and the conclusions stem clearly from data and discussion. References are coherent with the text.

Some minor issues come to mind:

- "Currently, computed tomography (CT) scan with contrast of chest was used to simulate heart compression fraction (HCF), which is the proportion of the heart compressed by CC [15], and HCF was subsequently mentioned in several studies to demonstrate high-quality CC during CPR in various populations [16,17]" this section could be expanded a little to explain clearly the link between heart compression fraction and high quality CPR.

- However this has been correctly addressed by the authors, it is to be noted that the population enrolled in the study may not be representative of the general population nor of the population most affected by cardiac arrest in Thailand (most notably only 3.3% of patients were admitted for cardiac illness). If there were any patients affected by cardiac arrest in the population in study, it may be reasonable to underline this data (otherwiseif patients resuscitated form cardiac arrest were excluded, absent or very few, then it should be disclosed as it is a significant limitation).

- The notion that "although lower HCF was found in the obese groups, sufficient HCF was generated at 60 mm CC depth" may be included in the conclusions, since it may yeald a relevant message for the local physicians to keep in mind when managing patients with higher BMI during cardiac arrest.

Other than these minor issues, the article seems fit for publication.

6. PLOS authors have the option to publish the peer review history of their article (what does this mean?). If published, this will include your full peer review and any attached files.

Reviewer #1: No

Reviewer #2: No

---

## [Author Response · Author response to Decision Letter 0]

19 Aug 2022

Dear Editor,

 Thank you very much for giving us the opportunity to revise our manuscript. We have responded to all questions and comments as described below; if any further unclear, please do let us know. 

Sincerely, 

Phatthranit Phattharapornjaroen

Corresponding author

Response to reviewers

#1. Is the manuscript technically sound, and do the data support the conclusions?

Reviewer #1: Partly

Reviewer #2: Yes

Answer: To add more information to our study, we have included sample size estimation and further statistical analysis parts as described on pages 6,7 Line 117-130. In addition, we have also added more conclusions and suggestions which relevant to the objective and our result on pages 8, lines 156-157, and on pages 10, lines 199-201. 

#2. Has the statistical analysis been performed appropriately and rigorously? 

Reviewer #1: No

Reviewer #2: Yes

Answer: For more rigorous statistical analysis, we performed further analysis by controlling for possible confounders (i.e., age and sex), which might affect the primary outcomes. Revised statistical part and results were on page 8, lines 156-157, and Table 4., subsequently. 

#3. Have the authors made all data underlying the findings in their manuscript fully available?

The PLOS Data policy requires authors to make all data underlying the findings described in their manuscript fully available without restriction, with rare exceptions (please refer to the Data Availability Statement in the manuscript PDF file). The data should be provided as part of the manuscript or its supporting information, or deposited to a public repository. For example, in addition to summary statistics, the data points behind means, medians and variance measures should be available. If there are restrictions on publicly sharing data—e.g. participant privacy or use of data from a third party—those must be specified.

Reviewer #1: No

Reviewer #2: Yes

Answer: We have provided the raw data in the supplementary file attached to this revised submission. 

4. Is the manuscript presented in an intelligible fashion and written in standard English?

Reviewer #1: Yes

Reviewer #2: Yes

Answer: Thank you for the reviewers’ comments.

#5. Review Comments to the Author

Reviewer #1: The depth of chest compressions depends on the rescuer's BMI anywhere in the world. My brother-in-law (Thai nurse) weighs 78 kg and 174 centimeters tall, he certainly has no problem make chest compression 5 -6 centimeters deep a European patient. Much depends on the difference in weight between the rescuer and the victim also in the Caucasians and

Asians populations.

Answer: We agreed with your comments that the factor associated with chest compression depth was the rescuers’ BMI. And there were several studies mentioned on this topic [1]. However, our research question focused on whether the recommendation depth from the USA and Europe regions were appropriate for Asian patients, who may have different sizes of anatomical parameters. In addition, we examined whether too deep or too shallow compression affects the quality of compression or internal organ injury.

References

Lin C-C, Kuo C-W, Ng C-J, Li W-C, Weng Y-M, Chen J-C. Rescuer factors predict high-quality CPR—a manikin-based study of health care providers. Am J Emerg Med. 2016;34: 20–24. doi:10.1016/j.ajem.2015.09.001

#Reviewer #2: This is an interesting retrospective study regarding the distribution of heart compression fraction among different sex and BMI in Thai population. The conclusion is that the recommended depth of chest compression during cardiac arrest (5-6 cm) is validated (with the simulated chest ct scan model in object) also in Thai population.

The article is well written in sound English, easy to read, and the conclusions stem clearly from data and discussion. References are coherent with the text.

Answer: Thank you for the reviewer’s comments and suggestions to improve the article.

#Some minor issues come to mind:

- "Currently, computed tomography (CT) scan with contrast of chest was used to simulate heart compression fraction (HCF), which is the proportion of the heart compressed by CC [15], and HCF was subsequently mentioned in several studies to demonstrate high-quality CC during CPR in various populations [16,17]" this section could be expanded a little to explain clearly the link between heart compression fraction and high quality CPR.

Answers According to the reviewer’s suggestion, we have added more information on page 4, lines 64-69

#- However this has been correctly addressed by the authors, it is to be noted that the population enrolled in the study may not be representative of the general population nor of the population most affected by cardiac arrest in Thailand (most notably only 3.3% of patients were admitted for cardiac illness). If there were any patients affected by cardiac arrest in the population in study, it may be reasonable to underline this data (otherwiseif patients resuscitated form cardiac arrest were excluded, absent or very few, then it should be disclosed as it is a significant limitation).

Answers We agree with the reviewer’s comment, and we have added this limitation on pages 9,10, lines 188-192.

#- The notion that "although lower HCF was found in the obese groups, sufficient HCF was generated at 60 mm CC depth" may be included in the conclusions, since it may yeald a relevant message for the local physicians to keep in mind when managing patients with higher BMI during cardiac arrest.

Answers We have added this information on page 10, lines 197-199.

---

## [Decision Letter · Decision Letter 1]

1 Dec 2022

Using Computed Tomography to Evaluate Proper Chest Compression Depth for Cardiopulmonary Resuscitation in Thai Population: A Retrospective Cross-Sectional Study

PONE-D-22-06863R1

Dear Dr. Phattharapornjaroen,

We’re pleased to inform you that your manuscript has been judged scientifically suitable for publication and will be formally accepted for publication once it meets all outstanding technical requirements.

Kind regards,

Simone Savastano

Academic Editor

PLOS ONE

Additional Editor Comments (optional):

Thank you very much for your work and for having addressed the reviewers' comments. Even though some scepticism of one of the reviewer I think that the paper might be published.

Reviewers' comments:

Reviewer's Responses to Questions

**Comments to the Author**

1. If the authors have adequately addressed your comments raised in a previous round of review and you feel that this manuscript is now acceptable for publication, you may indicate that here to bypass the “Comments to the Author” section, enter your conflict of interest statement in the “Confidential to Editor” section, and submit your "Accept" recommendation.

Reviewer #1: All comments have been addressed

Reviewer #2: (No Response)

2. Is the manuscript technically sound, and do the data support the conclusions?

Reviewer #1: Partly

Reviewer #2: Yes

3. Has the statistical analysis been performed appropriately and rigorously? 

Reviewer #1: No

Reviewer #2: Yes

4. Have the authors made all data underlying the findings in their manuscript fully available?

Reviewer #1: No

Reviewer #2: Yes

5. Is the manuscript presented in an intelligible fashion and written in standard English?

Reviewer #1: No

Reviewer #2: Yes

6. Review Comments to the Author

Reviewer #1: We have already received studies from Asian countries that hypothesize differences in quality in chest compressions given by rescuers with a smaller than slight of build. The effectiveness of compressions is given by constant training, correction of body posture, stiffness of the rescuer's arms, use of the pelvis as a fulcrum.

Therefore, during courses aimed at the Asian population, instructors should work with simulators that give immediate feedback regarding the depth of compressions and work with the student to improve the technique.

Reviewer #2: Thank you for addressing my precedent concerns; however i still have two comments:

- the formula in line 114 is not visible any more, maybe because of some formatting error?

- I respectfuly disagree with you conclusion in line 193-195: chronic lung and heart disease in fact do increase their volumes and alter thoracic anatomy.

7. PLOS authors have the option to publish the peer review history of their article (what does this mean?). If published, this will include your full peer review and any attached files.

Reviewer #1: No

Reviewer #2: No

---

## [Editor Report · Acceptance letter]

4 Dec 2022

PONE-D-22-06863R1 

Using Computed Tomography to Evaluate Proper Chest Compression Depth for Cardiopulmonary Resuscitation in Thai Population: A Retrospective Cross-Sectional Study 

Dear Dr. Phattharapornjaroen:

I'm pleased to inform you that your manuscript has been deemed suitable for publication in PLOS ONE. Congratulations! Your manuscript is now with our production department. 

Kind regards, 

on behalf of

Dr. Simone Savastano 

Academic Editor

PLOS ONE